# Upper Tract Urothelial Carcinoma: A Narrative Review of Current Surveillance Strategies for Non-Metastatic Disease

**DOI:** 10.3390/cancers16010044

**Published:** 2023-12-20

**Authors:** Jakob Klemm, Kensuke Bekku, Mohammad Abufaraj, Ekaterina Laukhtina, Akihiro Matsukawa, Mehdi Kardoust Parizi, Pierre I. Karakiewicz, Shahrokh F. Shariat

**Affiliations:** 1Department of Urology, University Medical Center Hamburg-Eppendorf, 20251 Hamburg, Germany; 2Department of Urology, Comprehensive Cancer Center, Medical University of Vienna, 1090 Vienna, Austria; gmd421030@s.okayama-u.ac.jp (K.B.); dr.abufaraj@gmail.com (M.A.); katyalaukhtina@gmail.com (E.L.); akihiro.matsukawa@meduniwien.ac.at (A.M.); mehdi.kardoustparizi@meduniwien.ac.at (M.K.P.); shahrokh.shariat@meduniwien.ac.at (S.F.S.); 3Department of Urology, Okayama University Graduate School of Medicine, Dentistry and Pharmaceutical Sciences, Okayama 700-8525, Japan; 4Division of Urology, Department of Special Surgery, Jordan University Hospital, The University of Jordan, Amman 11733, Jordan; 5Institute for Urology and Reproductive Health, Sechenov University, 119991 Moscow, Russia; 6Department of Urology, Jikei University School of Medicine, Tokyo 105-8461, Japan; 7Cancer Prognostics and Health Outcomes Unit, University of Montreal Health Centre, Montreal, QC H2X 3E4, Canada; pierre.karakiewicz@umontreal.ca; 8Hourani Center for Applied Scientific Research, Al-Ahliyya Amman University, Amman 11942, Jordan; 9Karl Landsteiner Institute of Urology and Andrology, 1090 Vienna, Austria; 10Department of Urology, Weill Cornell Medical College, New York, NY 10065, USA; 11Department of Urology, University of Texas Southwestern, Dallas, TX 75390, USA; 12Department of Urology, Second Faculty of Medicine, Charles University, 252 50 Prague, Czech Republic

**Keywords:** upper tract urothelial carcinoma, urothelial carcinoma, surveillance, follow-up

## Abstract

**Simple Summary:**

Upper tract urinary carcinoma (UTUC) is a rare type of cancer affecting the urinary system. Patients with UTUC often undergo surgeries like kidney-sparing surgery or radical nephroureterectomy. However, even after treatment, there remains a risk of the cancer recurring in different parts of the body. This narrative review aims to better understand the frequency and locations of such recurrences, which is crucial for effectively monitoring patients after their initial treatment. Currently, there is limited information on the optimal methods for tracking patients post-surgery, and on how early detection of cancer, before the appearance of symptoms, might improve health outcomes. This article presents the most important current guideline recommendations and elucidates the evidence behind them. Exploring new imaging technologies and improving methods for assessing patient risk, can potentially lead to more personalized and effective monitoring plans in the near future.

**Abstract:**

Non-metastatic upper urinary tract carcinoma (UTUC) is a comparatively rare condition, typically managed with either kidney-sparing surgery (KSS) or radical nephroureterectomy (RNU). Irrespective of the chosen therapeutic modality, patients with UTUC remain at risk of recurrence in the bladder; in patients treated with KSS, the risk of recurrence is high in the remnant ipsilateral upper tract system but there is a low but existent risk in the contralateral system as well as in the chest and in the abdomen/pelvis. For patients treated with RNU for high-risk UTUC, the risk of recurrence in the chest, abdomen, and pelvis, as well as the contralateral UT, depends on the tumor stage, grade, and nodal status. Hence, implementing a risk-stratified, location-specific follow-up is indicated to ensure timely detection of cancer recurrence. However, there are no data on the type and frequency/schedule of follow-up or on the impact of the recurrence type and site on outcomes; indeed, it is not well known whether imaging-detected asymptomatic recurrences confer a better outcome than recurrences detected due to symptoms/signs. Novel imaging techniques and more precise risk stratification methods based on time-dependent probabilistic events hold significant promise for making a cost-efficient individualized, patient-centered, outcomes-oriented follow-up strategy possible. We show and discuss the follow-up protocols of the major urologic societies.

## 1. Introduction

Upper urinary tract carcinoma (UTUC) is a relatively rare malady, constituting approximately 5–10% of urothelial carcinomas [1]. The treatment of non-metastatic UTUC is based on risk stratification into low- vs. high-risk tumors, with preferred therapeutic options being kidney-sparing surgery (KSS) or radical nephroureterectomy (RNU) with perioperative platinum-based chemotherapy when possible and indicated [2]. Indeed, the current guidelines from the European Association of Urology (EAU), the American Urological Association (AUA), and the National Comprehensive Cancer Network (NCCN) recommend offering neoadjuvant or adjuvant chemotherapy to high-risk UTUC patients, as they have been shown to improve disease-free survival (DFS), at least in non-metastatic UTUC patients [2,3,4,5]. Recognizing the potential for metachronous bladder tumors, local recurrences, or distant metastases after RNU or KSS, the EAU, AUA, and NCCN guidelines advocate robust surveillance protocols for UTUC patients [2,3,4]. However, evidence on follow-up strategies after definitive treatment for non-metastatic UTUC is low. For this reason, this article aims to summarize the available evidence on surveillance after surgery for non-metastatic UTUC with curative intent.

## 2. Evidence Acquisition

We searched the PubMed database up to 1 October 2023 using pre-defined search criteria as follows: (Upper Tract Urothelial Carcinoma) OR (Renal Pelvic Urothelial Cancer) OR (Ureteral Urothelial Cancer) AND (Surveillance) OR (Follow-up). In addition, we assessed the references in the EAU, AUA, and NCCN guidelines.

## 3. Prognostic Factors for UTUC

Assessment of established prognostic factors can help better understand which patients are at risk for disease recurrence and/or progression after definitive surgical therapy with curative intent for non-metastatic UTUC. However, it is essential to recognize that UTUC and bladder urothelial carcinoma are distinct entities, characterized by unique clinical, pathological, practical, and molecular factors [6].

### 3.1. Patient Related Factors

Advanced age and worse performance status have been significantly associated with decreased CSS in several studies including a systematic review including all published articles until December 2014 (Hazard ratio (HR) 1.02) [7]. Studies including a SEER database UTUC cohort analysis revealed that stage, grade, age, and sex were significantly associated with Cancer-Specific Survival (CSS) in 9208 non-metastatic UTUC patients treated with RNU [8,9,10,11]. However, as opposed to bladder cancer, sex was not associated with CSS in UTUC patients according to individual studies and a meta-analysis of 39,759 UTUC patients (pooled HR 0.94, 95% confidence interval (CI) = 0.89–1.00) [11,12]. Moreover, being a smoker at diagnosis has been shown to increase the risk of recurrence and mortality. In a retrospective study involving 864 clinically non-metastatic UTUC patients treated with RNU, of whom 202 were identified as heavy long-term smokers, the study found that heavy long-term smoking was significantly associated with advanced disease, disease recurrence, and worse CSS [13].

Although Lynch syndrome patients are at higher risk of developing UTUC [14], high microsatellite instability (MSI), which is a screening tool for Lynch syndrome in UTUC patients, seems to be associated with better overall survival [15].

Furthermore, residing in Balkan endemic nephropathy (BEN) areas has been independently linked to an increased risk of bladder recurrence following RNU for UTUC, with patients from these regions experiencing higher rates compared to those outside BEN areas (HR 1.81; *p* = 0.01) [16]. Additionally, a propensity-matched survival analysis revealed that patients in Taiwan and China who have undergone kidney transplantation are at a higher risk of developing UTUC than those without such a transplant history [17].

### 3.2. Tumor Stage and Grade

Despite the insights gained from these patient-related prognostic factors, the TNM stage and tumor grade continue to be the most influential prognostic indicators for UTUC [8,9,18,19,20]. A large retrospective study, including 13,314 patients with primary UTUC in the Netherlands between 1993 and 2017, reported five-year CSSs of 86% (95% CI = 84–87), 70% (95% CI = 68–72), and 44% for Ta/Tis, T1-T2, and non-organ confined tumors, respectively [21]. Simultaneously, a survival analysis of 6826 patients who underwent RNU for non-metastatic UTUC from the SEER database revealed a decreasing five-year CSS with increasing T stage: 86% for T1 high-grade N0 disease, 78% for T2 N0 disease, 63% for T3 N0, and 39% for T4 N0 or any N1–3 disease [22].

Moreover, M0 UTUC patients with lymph node involvement experience very poor five-year OS rates of approximately 15–30% [23,24,25]. In addition, extracapsular extension and lymph node density have been reported to be strong predictors of survival outcomes in N+ UTUC patients [26].

### 3.3. Tumor Characteristics

Tumor location has been shown to be associated with worse outcomes in univariable analyses [27]. According to a systematic review of 14,895 RNU patients, ureteral tumor location has a negative impact on CSS compared to pelvicalyceal tumors after adjusting for covariates (pooled HR of 1.52, *p* < 0.001) [28]. Multifocality is another factor that has been associated with disease recurrence and CSS in 2492 RNU patients, of whom 590 had multifocal tumors (HR 1.43, *p* = 0.019 and HR 1.46, *p* = 0.027, respectively) [29]. Furthermore, clinical tumor size has been linked to T stage, as shown by a large multi-institutional retrospective study of 932 RNUs for non-metastatic UTUC. The study demonstrated that a tumor size of 2 cm was the optimal cutoff to identify patients at risk for >T2 disease (decision curve analysis: clinical net benefit of 0.09 and a net reduction of 8 per 100 patients) [30]. An analysis of 4657 patients from the SEER database confirmed these findings: each 1 cm increase in tumor size translated into an adjusted odds ratio of 1.25 (*p* > 0.001) [31].

### 3.4. Other Pathological Features

Similarly to bladder UC, histological subtypes including micropapillary, squamous, and/or sarcomatoid were associated with worse CSS in a systematic review including 12,865 UTUC patients (pooled HR 2.00, 95% CI = 1.57–2.56) [32]. Additionally, lymphovascular invasion (LVI) was demonstrated to be an independent predictor of CSS in a large multicenter series including 763 UTUC patients who underwent RNU without neoadjuvant chemotherapy (HR 3.3, *p* = 0.005) [33]. Finally, as with other malignancies, positive surgical margins are associated with an increased risk of disease recurrence after RNU (HR 2.7, *p* = 0.001) [34]. 

In summary, risk factors serve as key indicators in forecasting the likelihood for a patient to experience recurrence of a disease, and, as such, they are integral to risk stratification [35], which in turn can help in patient counseling and shared decision-making based on evidence regarding intensification or deintensification of adjuvant therapy and surveillance. Indeed, prognostic tools integrating the above risk factors are the data-driven backbone to the development of effective and cost-serving surveillance protocols.

## 4. Current Surveillance Protocols after RNU

Leading urological and oncological associations such as the NCCN, AUA, and EAU propose surveillance protocols for both KSS- and RNU-treated patients. These protocols typically encompass a combination of regular cystoscopy, cytology, and imaging (Table 1, Table 2 and Table 3) [2,3,4]. The specific follow-up protocol for UTUC is generally determined based on the risk stratification group and the type of definitive therapy performed (i.e., KSS or RNU). Table 1, Table 2 and Table 3 provide an overview of the exact surveillance protocols of the EAU, AUA, and NCCN guidelines. Although little evidence exists on the value of these follow-up protocols, there is a rationale behind each of the surveillance modalities and regimens. In general, evidence shows that patients with asymptomatically detected recurrence have better overall survival, CSS, and recurrence-free survival than symptomatic UTUC patients [36].

### 4.1. Bladder Recurrence

Since bladder recurrence is common after definitive treatment for UTUC, cystoscopy is an integral part of the follow-up. The risk of bladder recurrence was assessed by two studies that created a nomogram to predict bladder recurrence at different time points. The UTUC collaboration group analyzed 1839 UTUC patients treated with open or laparoscopic RNU and found an intravesical recurrence in 31% of the patients with a median follow-up of 45 months [19]. Similarly, a retrospective study from Japan showed an intravesical recurrence rate of 29% after 5 years in 754 UTUC patients treated with RNU [37]. Nonetheless, no specific thresholds were found regarding intravesical recurrence risk and endoscopic surveillance. 

Conditional survival analyses take into account that a patient’s likelihood of bladder recurrence decreases with increasing recurrence-free survival. For example, Shigeta et al. analyzed 364 UTUC patients treated with open or laparoscopic RNU and found a 5-year intravesical recurrence-free survival (IVRFS) rate of 41.5%. However, the 5-year conditional IVRFS rate after 4 years of survivorship was 96.7% [38]. Similarly, the 5-year IVRFS rate of 3544 UTUC patients who underwent RNU was 55% but the conditional 5-year IVRFS rate increased to 90% after 4 years of survivorship [39]. These data support that the guideline recommendations cover a timeframe of 5 years. However, the retrospective study design and the relatively small patient cohort of the Shigeta study represent major limitations, meaning that no clear conclusions can be drawn regarding endoscopic surveillance protocols. 

Martini et al. recommended continuing cystoscopy follow-up for more than 10 years, especially for patients with a prior history of bladder cancer. This recommendation is supported by Weibull regression models for the hazard rate of recurrence and other-cause mortality, which indicate a higher risk of recurrence than other-cause mortality for patients under 70 years of age [40]. The impact of a single dose of intravesical post-operative chemotherapy lowers the risk of intravesical recurrence [41] and could lead to a de-escalation of cystoscopies, but it is unclear to what extent. After KSS, the rate of IVR is likely to be higher due to many interventions with ureteroscopies, which have been shown to lead to higher rate of IVR due to presumed seeding [42].

### 4.2. Local and Distant Recurrence

Unlike intravesical recurrences, which can be effectively monitored through frequent cystoscopies, the emergence of loco-regional and distant recurrences necessitates regular abdominopelvic and chest imaging. Generally, local recurrence rates after RNU are reported to range from 5 to 32% in retrospective studies [43,44,45,46,47,48]. In a study conducted by Martini et al., the risk of non-bladder recurrence was assessed in 1378 prospectively collected UTUC patients treated with RNU across various European academic centers. Patients were classified into two groups based on their prior history of bladder cancer. After 2 years, the risk of non-bladder recurrence was 42% for patients without and 47% for those with a history of prior bladder cancer. Considering that European guidelines advocate a deintensification of imaging after 2 years, the authors suggest maintaining a semiannual imaging schedule until after the fourth year when the risk curve for non-bladder recurrence post RNU significantly plateaus [40]. 

A study from Japan that followed 733 UTUC patients post RNU found a non-bladder recurrence rate of 34% within 5 years, with most recurrences occurring within the first 3 years following treatment [49]. The study also indicated a correlation between the location of the primary tumor and patterns of recurrence or metastasis. Lower and middle ureter tumors were more prone to local recurrence in the pelvic cavity, while tumors in the renal pelvis or upper ureter were more likely to metastasize to the lungs or liver [49]. 

In terms of metastatic patterns in UTUC patients, primary metastatic UTUC patients are most likely to experience lung (36.1%) bone (27%), or liver (19.1%) metastasis, with brain metastasis only in 1.6% of cases (SEER: 9436 primary UTUC patients) [50]. These patterns were corroborated by Tanaka et al., who found that 30% of 733 low- and high-risk UTUC patients experienced recurrence within 3 years post RNU, with distant recurrences accounting for 56% of these cases. The predominant sites of metastasis were the lungs, liver, and bones [49]. Other studies have reported similar metastasis rates post RNU, ranging from 8.3% to 46% [44,45,46,47,48,51]. Consequently, abdominal and chest imaging are part of the surveillance protocols of the NCCN, AUA, and EAU guidelines, allowing for the detection of these metastasis patterns. 

### 4.3. Risk Stratification and Surveillance

Risk stratification plays a crucial role in striking the right equilibrium between the intensity and frequency of surveillance modalities tailored for each patient. A retrospective analysis of 1029 UTUC patients who underwent RNU in Canada revealed that 73% of patients experienced recurrence at any site (urothelial, local, and distant recurrences) within the first two years [52]. Based on these findings, the authors suggested a surveillance protocol based on three risk groups: low risk (pTa-T1, pN0, low grade, no LVI, unifocality), intermediate risk (pTa-T1, pN0, +/− high grade, +/− LVI, +/− multifocality), and high risk (≥pT2 and/or pN+), with the first two years post RNU demanding heightened surveillance, particularly for high-risk patients [52]. 

In a retrospective analysis of 426 UTUC patients who underwent RNU, Momota et al. sought to assess the cost-effectiveness of surveillance protocols. Initially, a pathology-based surveillance protocol (normal risk: ≤pT2N0; high risk: N0 with pT3 or LVI+; very high risk: pT4, positive surgical margin, or lymph node involvement (cN+ or pN+)) was utilized; however, it fell short in its ability to effectively distinguish between patients with a high risk of recurrence and those without. The authors subsequently improved cost-effectiveness by implementing a risk-score-based surveillance protocol which weighted different risk factors, resulting in a 55% reduction in the cost of 5-year surveillance [53]. However, it should be noted that the authors did not include grading in their risk stratification, resulting in risk groups with mixed grading. 

In a retrospective analysis of 714 UTUC patients who underwent RNU, Shigeta et al. found that smokers had a higher risk of UTUC-related death compared to non-smokers according to Weibull model estimates. The authors suggest that extending surveillance may be necessary for this population to detect and manage potential recurrences or metastases [54]. Yet, similar to the Momota study, this study also had a major limitation in risk stratification, as it was not based on the standard pathological features or risk groups recommended by guidelines.

The key questions are the optimal protocol and the duration of surveillance. Lindner et al. studied time-to-tumor recurrence in 54 UTUC patients post RNU and 14 UTUC patients post KSS. They discovered that 38.9% of patients post RNU developed distant metastasis, with the vast majority (85.7%) occurring within the first year post surgery. Only 9.5% and 4.8% occurred in the second and third years, respectively [55]. These findings underscore the rationale of the EAU guidelines’ recommendation to de-intensify imaging after the first two years post RNU.

Finally, the delivery of adjuvant systemic therapy with platinum-based combination has shown to lower recurrence rates and may impact the intensity of surveillance imaging [56].

In conclusion, considering their elevated risk of recurrence, patients with UTUC post RNU necessitate a meticulous follow-up regimen. This should encompass routine chest and abdominopelvic imaging, supplemented with periodic cystoscopy. It is particularly crucial to maintain an intensified monitoring schedule during the initial two years of surveillance.

## 5. Current Surveillance Protocols after Endoscopic Treatment

A kidney-sparing approach is generally recommended to reduce morbidity associated with radical surgery [2] for UTUC tumors with low-risk features including all of the following: unifocal disease, tumor size < 2 cm, negative cytology, low-grade ureterorenoscopy (URS) biopsy, and no invasive aspect on computed tomography (CT). Current surveillance strategies for UTUC patients after KSS (either for low-risk tumors or for imperative indications such as solitary kidney, bilateral UTUC, chronic kidney disease, or any other comorbidity compromising the use of RNU) include imaging and periodic cystoscopy but also URS as a standard (Table 1, Table 2 and Table 3) [2]. 

### 5.1. Recurrence Rates

The design of surveillance strategies is fundamentally shaped by recurrence rates and patterns. This principle is illustrated in a recent systematic review that scrutinized the oncologic outcome of endoscopic surgeries for UTUC including 1091 patients from twenty studies with mostly low-grade tumors. The authors found a pooled bladder recurrence rate of 35% (95% CI 28–42.3: I^2^ = 48%) after retrograde URS and 17.7% (95% CI 6.5–32.1: I^2^ = 29%) after antegrade URS. Additionally, the pooled rate of upper urinary tract recurrence was 56.4% (95% CI 41.2–70.9: I^2^ = 93%) after retrograde treatment and 36.2% (95% CI 25.5–47.6: I^2^ = 57%) after antegrade treatment [57].

Interestingly, Mohapatra et al. retrospectively analyzed patients who initially received endoscopic treatment at two large tertiary referral centers in the U.S. for low- and high-risk UTUC and found that 80% of their cohort experienced disease progression to high-risk UTUC, triggering an RNU within 5 years [58].

Given these results, it is crucial to implement a rigorous follow-up schedule involving repeated cystoscopy and ureteroscopy, particularly after endoscopic treatment. This strategy ensures timely detection of any disease recurrence or progression, and keeps the option for RNU open, preserving the critical window of opportunity for intervention if required.

### 5.2. Cytology

Urinary cytology serves as a crucial tool for diagnosing UTUC and its subsequent surveillance, particularly in high-risk tumors, regardless of the definitive therapy approach. This tool is widely recommended across various guidelines due to its high specificity and non-invasive nature [2,3,4]. The adoption of the Paris System for Reporting Urinary Cytology (TPS) has led to varying results with respect to sensitivity and specificity. Studies have reported sensitivity ranging from 19% to 82%, while specificity ranges from 86% to 100% for primary diagnosis [59]. Interestingly, these results align with a meta-analysis from the pre-TPS era, which demonstrated a pooled sensitivity of 53% (95% CI = 42–64; I^2^ = 86%) and a pooled specificity of 90% (95% CI = 85–93; I^2^ = 0%) for UTUC detection in upper urinary tract cytology during primary diagnosis [60].

Given these findings, cytology is also recommended for UTUC follow-up. However, it is important to note that the impact of the Paris System on sensitivity and specificity during surveillance is yet to be determined.

### 5.3. Second-Look URS and Endoscopic Follow-Up

EAU and AUA guidelines agree on an early repeated URS which should be performed one to three and six months after KSS [2,3]. Villa et al. demonstrated a high cancer detection rate of 51.2% during the second URS 6–8 weeks after initial URS treatment in 41 UTUC patients with high- or low-risk UTUC. Patients who had a positive result during the second URS had an 81.3% likelihood of also having a positive result during the third URS, in contrast to a cancer detection rate of 41.2% for the third URS following a negative second URS result [61]. The authors therefore strongly recommend a second-look URS within 6–8 weeks after initial endoscopic management. 

Several studies provide further evidence for the importance of close endoscopic follow-up in UTUC patients who undergo KSS. Kawada et al. conducted a systematic review and found that endoscopically managed tumors had similar oncologic outcomes to those managed with RNU, but recurrence in the upper urothelial tract was observed in 28–85% of patients across the studies [62]. Lindner et al. reported a stable recurrence rate after KSS (endoscopic treatment and ureterectomy) at 12.5% to 20.5% per year during the first 5 years after surgery, with six upper urinary tract recurrences, two bladder recurrences, and two lymph node and/or distant metastases in 14 KSS patients (57.1% high-grade tumors). As most of the recurrences after KSS concerned the bladder or upper urinary tract (six upper urinary tract recurrences, two bladder recurrences), the authors suggest, contrary to the current EAU, AUA, and NCCN guidelines, that cystoscopies and URS should not be deintensified after 2 years of follow-up [55].

This is further emphasized by Seisen et al., who analyzed 42 primary endoscopically treated patients with mostly low-grade tumors. The local recurrence-free survival defined as recurrence in the operation site was 35.7% after endoscopic treatment. Additionally, Kaplan–Meier curves showed a consistent incidence of local recurrence over the 5-year follow-up period [63]. Moreover, Hendriks et al. found a higher IVR rate for high-risk UTUC patients treated with KSS (endoscopic treatment and ureterectomy) compared to RNU in a propensity-score-matched cohort based on EAU risk stratification (52% for KSS vs. 32% for RNU; *p* = 0.029) [64]. 

In conclusion, while the existing evidence may be of a low level, it, nonetheless, underscores the necessity for rigorous surveillance, including URS, following the endoscopic management of UTUC.

Specifically, because KSS is an alternative to RNU, it has promise to be safe while retaining the renal unit.

## 6. Current Surveillance Protocols after Segmental or Distal Ureterectomy

Segmental and distal ureterectomy represent non-endoscopic kidney-sparing approaches that are recommended for low-risk UTUC tumors [2]. Seisen et al. assessed oncologic outcomes of ureterectomies compared with RNU in a systematic review including 586 segmental ureterectomy patients from comparative studies and reported no differences in CSS, OS, and RFS between the two groups [65]. Seisen also showed that the 5-year local RFS, defined as recurrence in the operation site, ranged from 37% to 91% across studies [66,67,68,69]. Similarly, Fang et al. included 983 ureterectomy patients from comparative studies and reported no differences in oncologic outcomes between ureterectomy and RNU (CSS: HR 0.90, *p* = 0.33, OS: HR 0.98, *p* = 0.93, and RFS: HR 1.06, *p* = 0.72). However, patients undergoing ureterectomy were more likely to harbor favorable pathological features. The cumulative recurrence rate 5 years after surgery ranged from 16% to 72% across three studies comprising 165 patients [66,68,70,71]. 

Due to ureteroureterostomy or ureteroneocystostomy, which complicate endoscopic follow-up, surveillance after ureterectomy was frequently based on the follow-up regimen of RNUs across retrospective studies [66,68]. Yet, Kim et al. retrospectively analyzed 394 RNU patients and 44 segmental ureterectomy patients and found no significant differences regarding 3-year PFS and IVRFS (68% vs. 73%: *p* = 0.9 and 42% vs. 37%: *p* = 0.8, respectively). Interestingly, the authors stated the use of semiannual ureteroscopy in their follow-up regimen for patients treated with segmental ureterectomy [72]. While ureterectomies are considered kidney-sparing approaches for UTUC, the surveillance protocols recommended by guidelines do not distinguish between endoscopic management and ureterectomy, leading to non-compliance with guideline protocols, especially after ureterectomy. As of now, this issue has not been addressed in the guidelines. 

## 7. Discussion

The existing guideline recommendations for surveillance following definitive therapy for UTUC largely rely on expert opinions and low-level evidence due to the relative rarity of this disease, which has led to a dearth of large prospective randomized clinical trials that could bolster the evidence. The available data frequently hail from retrospective studies involving small cohorts. Most existing data on surveillance, recurrence, and progression center around RNU. Given that UTUC patients, particularly those in the high-risk category, are generally at substantial risk of recurrence or progression, frequent follow-up examinations seem necessary. 

The challenge for healthcare providers lies in delivering an appropriate surveillance protocol with the correct frequency to suit each individual patient and tumor type. As highlighted by Momota et al., risk stratification plays a pivotal role in selecting the appropriate patient and surveillance protocol, and it significantly impacts healthcare system costs, resource utilization, and patient convenience/quality of life [53]. Currently, risk stratification largely depends on tumor T and N stage as well as grade. However, the advent of next-generation sequencing (NGS) in recent years has led to the discovery of molecular subtypes of various tumor entities, which are gradually being incorporated into clinical practice. Additionally, liquid biopsies, which utilize blood or other body fluids like urine, are being explored and show promise in the detection of UTUC [73]. The utilization of biomarkers, in general, has the potential to enhance surveillance protocols. The integration of biomarkers and liquid biopsies could lead to less-invasive disease monitoring, improved detection accuracy, and the identification of novel therapeutic targets, thereby potentially reducing the risk of recurrence. Incorporating such biomarkers into routine surveillance could transform patient management, allowing for more personalized and effective monitoring strategies.

A systematic review assessing UTUC alterations revealed significant differences between UTUC and urothelial bladder cancer, particularly in areas such as activated FGFR3 signaling, the extent of altered somatic expression of DNA mismatch repair genes, and individual UTUC molecular subtypes [74]. The impact of these discoveries on future treatments and, consequently, future surveillance protocols in UTUC is in its burgeoning phase and much is expected.

Furthermore, emerging urinary biomarkers might offer the potential to reduce patient discomfort by providing equivalent diagnostic value without necessitating invasive ureterorenoscopy during follow-up. Territo et al. evaluated the diagnostic worth of EpiCheck, a urine test based on the analysis of 15 DNA methylation biomarkers, and the results were promising with a sensitivity/negative predictive value (NPV) for high-grade tumors of 96%/97%, compared to 71%/86% for cytology [75]. However, additional evidence is required to integrate these novel urinary biomarkers into surveillance protocols.

Additionally, by enhancing precise imaging techniques, patients can not only be directed towards the treatment approach that best suits their individual needs, but also the accuracy of risk stratification can be improved. This enhancement may help circumvent unnecessary and invasive surveillance methods such as URS. Moreover, advancements in imaging such as PET/CT and PET/MRI, which seem to be promising imaging tools for the detection of lymph nodes and distant metastases in urothelial bladder cancer, hold the potential to change surveillance protocols and risk stratification by offering a more accurate nodal and distant assessment of the disease [76].

## 8. Conclusions

The evidence supporting surveillance protocols following definitive therapy for UTUC is currently sparse, and predominantly reliant on low-level evidence and expert opinion. Given the rarity of UTUC, conducting large prospective randomized clinical trials may prove challenging, underscoring the need for refined risk stratification methods. Surveillance protocols may be optimized in the future to meet the individual needs of each patient by enhancing risk stratification accuracy through more precise imaging and/or the implementation of novel urine and blood-based biomarkers.

## Figures and Tables

**Table 1 cancers-16-00044-t001:** Displaying the current EAU guideline surveillance protocol.

**EAU**	Months	3	6	9	12	15	18	21	24	30	36	42	48	54	60	
**low risk after RNU**	cytology	not mandatory														
	cystoscopy	◯			◯				◯		◯		◯		◯	
	CT/MR urography	not mandatory														
**high risk after RNU**	cytology	◯	◯	◯	◯	◯	◯	◯	◯	◯	◯	◯	◯	◯	◯	annually thereafter
	cystoscopy	◯	◯	◯	◯	◯	◯	◯	◯	◯	◯	◯	◯	◯	◯	annually thereafter
	CT/MR urography		◯		◯		◯		◯		◯		◯		◯	annually thereafter
	Chest CT		◯		◯		◯		◯							
**low risk after KSS**	cytology															
	cystoscopy	◯	◯		◯				◯		◯		◯		◯	
	CT/MR urography	◯	◯		◯				◯		◯		◯		◯	
	URS	◯														
**high risk after KSS**	cytology	◯	◯													
	cystoscopy															
	CT/MR urography															
	URS	◯	◯													
**low risk**	unifocal, tumor size < 2 cm, low-grade cytology, low-grade URS biopsy, no invasive aspect on CT urography (all of these)															
**high risk**	hydronephrosis, tumor size ≥ 2 cm, high-grade cytology, high-grade URS biopsy, multifocal, previous RC for MIBC, variant histology (any of these)															

CT = computed tomography; MR = magnet resonance; URS = ureterorenoscopy. ◯ = recommended.

**Table 2 cancers-16-00044-t002:** Displaying the current AUA guideline surveillance protocol.

**AUA**	Months	3	6	9	12	15	18	21	24	30	36	42	48	54	60
**<pT2 N0/M0 after RNU**	cystoscopy, cytology	◯	◯		◯		◯		◯		◯		◯		◯
	cross-sectional imaging *		◯		◯		◯		◯		◯		x		x
	chest imaging				◯										
	BMP **	◯			◯		◯		◯		◯		◯		◯
**>pT2 Nx/0 after RNU**	cystoscopy, cytology	◯	◯		◯		◯		◯	◯	◯		◯		◯
	cross-sectional imaging *	◯	◯		◯		◯		◯		◯		◯		◯
	chest imaging	◯	◯		◯		◯		◯		◯		◯		◯
	BMP **	◯			◯		◯		◯		◯		◯		◯
**low risk after KSS**	cystoscopy, cytology	◯	◯		◯		◯		◯		◯		◯		◯
	URS	◯	◯		◯										
	cross-sectional imaging *	◯			◯		◯		◯		◯		◯		◯
	chest imaging				◯										
	BMP **				◯				◯		◯		◯		◯
**high risk after KSS**	cystoscopy, cytology	◯	◯		◯		◯		◯	◯	◯		◯		◯
	URS	◯	◯		◯										
	cross-sectional imaging *	◯	◯		◯		◯		◯	◯	◯		◯		◯
	chest imaging		◯		◯		◯		◯		◯				
	BMP **				◯				◯		◯		◯		◯
**low risk**	Bx: low-grade; cytology: no HGUC, <cT2 N0M0, no sessile or flat tumors														
**high risk**	BX: high-grade; cytology: HGUC, ≥cT2 N+														

* Cross-sectional imaging of the abdomen and pelvis should be performed with contrast when possible; ** basic metabolic panel. ◯ = recommended; x = optional.

**Table 3 cancers-16-00044-t003:** Displaying the current NCCN guideline surveillance protocol.

**NCCN**		3	6	9	12	15	18	21	24	30	36	42	48	54	60	months
**pT0–1 after RNU**	cytology	◯	◯	◯	◯	longer intervals not specified										
	cystoscopy	◯	◯	◯	◯											
	cross-sectional imaging *	not specifically recommended														
**pT2–4, pN+ after RNU**	cytology	◯	◯	◯	◯	longer intervals not specified										
	cystoscopy	◯	◯	◯	◯											
	cross-sectional imaging *	not specifically recommended														
	chest imaging															
**pT0–1 after KSS**	cytology	◯	◯	◯	◯	longer intervals not specified										
	cystoscopy	◯	◯	◯	◯											
	cross-sectional imaging *	3–12-month intervals														
	URS															
**pT2–4, pN+ after KSS**	cytology	◯	◯	◯	◯	longer intervals not specified										
	cystoscopy	◯	◯	◯	◯											
	cross-sectional imaging *	3–12-month intervals														
	chest imaging															
	URS															

* Abdominal/pelvic CT or MRI with and without contrast. ◯ = recommended.

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
