# Peer review of "Upper Tract Urothelial Carcinoma: A Narrative Review of Current Surveillance Strategies for Non-Metastatic Disease"

_cancers, 2023, doi:10.3390/cancers16010044_

Round 1

Reviewer 1 Report

Comments and Suggestions for Authors

Regarding the surveillance of postoperative UTUC, the current situation, problems, and future challenges are thoroughly described, based on guidelines and much literature. It is well-organized to aid the understanding of readers specializing in urinary cancers and serves as a reference for routine medical care of UTUC. Although the main text feels like to be a little wordy, I get the impression that the main points are written in this article. If possible, the discussion could include more examples of biomarkers that look promising for UTUC. I think it would have been nice to have some comments about how surveillance strategies would change if we had biomarkers.

Comments on the Quality of English Language

As commented above, this is a topic for clinicians who are struggling with post-operative surveillance for UTUC. Although there is no answer written to our struggles, I believe that this will be helpful for clinicians when conducting postoperative surveillance of individual UTUC patients. Honestly speaking, the content is not particularly impressive, but clinicians specializing in urinary cancer consider it to be a meaningful paper.

Author Response

Reviewer #1:

Regarding the surveillance of postoperative UTUC, the current situation, problems, and future challenges are thoroughly described, based on guidelines and much literature. It is well-organized to aid the understanding of readers specializing in urinary cancers and serves as a reference for routine medical care of UTUC. Although the main text feels like to be a little wordy, I get the impression that the main points are written in this article. If possible, the discussion could include more examples of biomarkers that look promising for UTUC. I think it would have been nice to have some comments about how surveillance strategies would change if we had biomarkers.

Thank you for the time you have dedicated to reviewing our manuscript and for your insightful comment. In response to your suggestion regarding the inclusion of more details on biomarkers and their potential impact on UTUC surveillance strategies, we have expanded the discussion as follows:

„Additionally, liquid biopsies, which utilize blood or other bodily fluids like urine, are being explored and show promise in the detection of UTUC(73). The utilization of biomarkers, in general, has the potential to enhance surveillance protocols. The integration of biomarkers and liquid biopsies could lead to less invasive disease monitoring, improved detection accuracy, and the identification of novel therapeutic targets, thereby potentially reducing the risk of recurrence. Incorporating such biomarkers into routine surveillance could transform patient management, allowing for more personalized and effective monitoring strategies.“

As commented above, this is a topic for clinicians who are struggling with post-operative surveillance for UTUC. Although there is no answer written to our struggles, I believe that this will be helpful for clinicians when conducting postoperative surveillance of individual UTUC patients. Honestly speaking, the content is not particularly impressive, but clinicians specializing in urinary cancer consider it to be a meaningful paper.

Thank you for your honest feedback. We understand that while our manuscript may not provide groundbreaking new insights, it was crafted with the goal of summarizing existing evidence and offering a comprehensive overview for clinicians managing post-operative surveillance of UTUC patients. We are gratified to know that you think that specialists in urinary cancer find it to be a meaningful contribution. Our aim was to consolidate relevant information in a clear and accessible format, thereby aiding clinicians in informed decision-making for individual patient care in this complex field.

Reviewer 2 Report

Comments and Suggestions for Authors

The authors provide a comprehensive and informative review of surveillance strategies for non-metastatic upper urinary tract carcinoma (UTUC).  The inclusion of surveillance protocols recommended by major urologic societies such as the EAU and AUA is particularly valuable for clinicians. However, there are some areas that need improvement.

1. In table 1, what is the surveillance protocol for high-risk patients receiving KSS after 6 months?  

2. The manuscript could benefit from a more detailed discussion of novel surveillance techniques such as liquid biopsy and their potential impact on UTUC surveillance.

Author Response

Reviewer #2:

The authors provide a comprehensive and informative review of surveillance strategies for non-metastatic upper urinary tract carcinoma (UTUC). The inclusion of surveillance protocols recommended by major urologic societies such as the EAU and AUA is particularly valuable for clinicians. However, there are some areas that need improvement.

Thank you very much for the time and effort you have invested in reviewing our manuscript. Your feedback is greatly appreciated.

  1. In table 1, what is the surveillance protocol for high-risk patients receiving KSS after 6 months?

Thank you for your thorough review of our tables. We acknowledge your query regarding the surveillance protocol for high-risk UTUC patients post-6 months as per the EAU guidelines. Regrettably, the EAU does not offer specific recommendations for the time frame after 6 months due to a paucity of data. Our table was designed to concisely present the existing, specific recommendations provided by the EAU.

  1. The manuscript could benefit from a more detailed discussion of novel surveillance techniques such as liquid biopsy and their potential impact on UTUC surveillance.

Thank you for your valuable suggestion. To address this, we have expanded the discussion section to include more detailed information on novel surveillance techniques like liquid biopsy and their potential impact on UTUC surveillance. Specifically, we added:

Additionally, liquid biopsies, which utilize blood or other body fluids like urine, are being explored and show promise in the detection of UTUC(73). The utilization of biomarkers, in general, has the potential to enhance surveillance protocols. The integration of biomarkers and liquid biopsies could lead to less invasive disease monitoring, improved detection accuracy, and the identification of novel therapeutic targets, thereby potentially reducing the risk of recurrence. Incorporating such biomarkers into routine surveillance could transform patient management, allowing for more personalized and effective monitoring strategies.”

Reviewer 3 Report

Comments and Suggestions for Authors

 It is a well-written reivew artilce, however some points should be reinforced and some should be corrected.

1. Some area or disease contition with high UTUC incidence rate such as Balkan endemic nephropathy and  Taiwanese post-kidney transplantation patients should be addressed in this review.

2. page 9 line 347  ... treated with segmental ""ureteroscopy "" ....    ureterosocpy may be the mis-typing of uretecotmy .

2.  

Author Response

Reviewer #3:

It is a well-written reivew artilce, however some points should be reinforced and some should be corrected.

First and foremost, we would like to extend our sincere thanks to the reviewer for the time and effort dedicated to reviewing our manuscript.

  1. Some area or disease contition with high UTUC incidence rate such as Balkan endemic nephropathy and Taiwanese post-kidney transplantation patients should be addressed in this review.

Thank you for emphasizing these critical areas. To adequately address these topics, we have expanded the prognostic factors section with the following additions:

“Furthermore, residing in Balkan endemic nephropathy (BEN) areas has been in-dependently linked to an increased risk of bladder recurrence following RNU for UTUC, with patients from these regions experiencing higher rates compared to those outside BEN areas (HR 1.81; p=0.01)(16). Additionally, a propensity-matched survival analysis re-vealed that patients in Taiwan and China who have undergone kidney transplantation are at a higher risk of developing UTUC than those without such a transplant history(17).”

  1. page 9 line 347 ... treated with segmental ""ureteroscopy "" .... ureterosocpy may be the mis-typing of uretecotmy

Thank you for your meticulous attention to detail in reviewing our manuscript. You are correct; the term should indeed be 'ureterectomy' instead of 'ureteroscopy.' We have amended the sentence accordingly to read:

“Interestingly, the authors stated to use a semiannually ureteroscopy in their follow-up regimen for patients treated with segmental ureterectomy.”
